# BH3 Mimetic Sensitivity of Colorectal Cancer Cell Lines in Correlation with Molecular Features Identifies Predictors of Response

**DOI:** 10.3390/ijms22083811

**Published:** 2021-04-07

**Authors:** Le Zhang, Prashanthi Ramesh, Maxime Steinmetz, Jan Paul Medema

**Affiliations:** 1Laboratory for Experimental Oncology and Radiobiology, Center for Experimental and Molecular Medicine, Cancer Center Amsterdam, Amsterdam UMC, University of Amsterdam, Meibergdreef 9, 1105 AZ Amsterdam, The Netherlands; l.zhang@amsterdamumc.nl (L.Z.); p.ramesh@amsterdamumc.nl (P.R.); m.z.steinmetz@amsterdamumc.nl (M.S.); 2Oncode Institute, Meibergdreef 9, 1105 AZ Amsterdam, The Netherlands

**Keywords:** colorectal cancer, BH3 mimetics, BCL-2, BCL-XL, MCL-1, consensus molecular subtype (CMS)

## Abstract

Colorectal cancer (CRC) is a heterogeneous disease, which in part explains the differential response to chemotherapy observed in the clinic. BH3 mimetics, which target anti-apoptotic BCL-2 family members, have shown potential in the treatment of hematological malignancies and offer promise for the treatment of solid tumors as well. To gain a comprehensive understanding of the response to BH3 mimetics in CRC and the underlying molecular factors predicting sensitivity, we screened a panel of CRC cell lines with four BH3 mimetics targeting distinct anti-apoptotic BCL-2 proteins. Treatment with compounds alone and in combination revealed potent efficacy of combined MCL-1 and BCL-XL inhibition in inducing CRC cell death, irrespective of molecular features. Importantly, expression of the anti-apoptotic protein target of BH3 mimetics on its own did not predict sensitivity. However, the analysis did identify consensus molecular subtype (CMS) specific response patterns, such as higher resistance to single and combined BCL-2 and MCL-1 inhibition in CMS2 cell lines. Furthermore, analysis of mutation status revealed that *KRAS* mutant cell lines were more resistant to MCL-1 inhibition. Conclusively, we find that CRC cell lines presented with distinct responses to BH3 mimetics that can in part be predicted by their CMS profile and *KRAS*/*BRAF* mutations. Overall, almost all CRC lines share sensitivity in the nanomolar range to combined MCL-1 and BCL-XL targeting suggesting that this would be the preferred approach to target these cancers.

## 1. Introduction

Despite significant advances in therapy, colorectal cancer (CRC) is still one of the leading causes of cancer-related deaths worldwide. CRC is characterized by high heterogeneity and unfavorable overall survival in late stage disease [1]. Due to the heterogeneity of the molecular and pathological phenotypes, CRC patients exhibit differential response to adjuvant therapies [2,3]. Based on the distinct molecular gene expression patterns, a consensus classifier was established to stratify CRC into four consensus molecular subtypes (CMS) to better understand the biological heterogeneity [4]. CMS1 displays high microsatellite instability (MSI), often associated with high immunogenicity. CMS2 is the classical subtype characterized by an epithelial phenotype harboring *WNT*/*MYC* activation. CMS3 tumors exhibit metabolic activation and are enriched for *KRAS* mutations, whereas CMS4 is a poor-prognosis mesenchymal subtype, which is characterized by an increased stromal infiltrate and chemotherapy resistance.

The BCL-2 protein family is a cluster of proteins regulating the intrinsic apoptosis pathway, which can be sub-classified into three subgroups according to their corresponding structure and function. Firstly, BCL-2-associated X protein (BAX), Bcl-2 homologous antagonist/killer (BAK), and Bcl-2-related ovarian killer protein (BOK), the effectors of apoptosis, are able to form pore-like structures on the outer membrane of mitochondria and facilitate the release of cytochrome c to unleash apoptotic cell death [5]. In the absence of an apoptotic signal, these effectors are kept in check by anti-apoptotic BCL-2 family members, including BCL-2, BCL-XL, BCL-W, and MCL-1. Anti-apoptotic BCL-2 proteins translocate to the mitochondrial outer membrane and block the activation of BAX/BAK/BOK [6]. Activation of the apoptotic cascade usually starts with the third group of BCL-2 family members, the BH3 only proteins. Their upregulation or activation and subsequent interaction with pro- and anti-apoptotic BCL-2 family members tips the balance in favor of apoptosis allowing the effectors to act [7]. Tumor cells frequently upregulate anti-apoptotic BCL-2 family members and/or downregulate pro-apoptotic members, conferring resistance to apoptotic signals including those induced by radiotherapy and chemotherapy [8]. In order to sensitize cancer cells to apoptosis, a group of bioactive compounds, so called BH3 mimetics, which functionally mimic intrinsic BH3-only proteins, were developed and show promising efficacy in inducing apoptosis in hematologic malignancies and solid tumors. Corresponding to their selective affinity, BH3 mimetics exhibit distinct specificity in targeting different anti-apoptotic BCL-2 proteins (Table 1). ABT-199 (Venetoclax) is a specific BCL-2 inhibitor and is FDA-approved for treatment of chronic lymphocytic leukemia (CLL) or small lymphocytic lymphoma (SLL). A-1155463 is a specific BCL-XL inhibitor that shows potent efficacy in several solid tumor models [9,10]. AZD5991, an MCL-1 specific inhibitor, is under investigation in a clinical trial for relapsed or refractory hematological malignancies [11] (ClinicalTrials.gov Identifier: NCT03218683). ABT-263 (Navitoclax), which targets BCL-2, BCL-XL, and BCL-W, is FDA approved as well for combination therapy in hematological malignancies and is being studied in multiple clinical trials for several solid tumors (ClinicalTrials.gov Identifier: NCT02591095; NCT00878449; NCT00891605). Of note, despite their remarkable efficacy in pre-clinical models, only six BH3 mimetics have entered clinical trials, partly due to on-target side-effects on platelets and hematopoietic cells. However, appropriate dosing strategies as well as targeted delivery in antibody-drug conjugate configurations that can overcome these toxicities are under investigation. Additionally, response to BH3 mimetics both in normal and cancer cells is also dependent on tissue origin and distinct molecular properties. Therefore, a better understanding of the anti-apoptotic dependencies in the context of the diverse molecular subtypes of CRC is needed to define the optimal strategy [12,13]. 

Herein, we tested four different BH3 mimetics targeting the different anti-apoptotic BCL-2 proteins in 19 CRC lines, which have been classified into four CMSs [14]. Our data show CMS2 lines to be relatively resistant to both BCL-2 or MCL-1 inhibition. In addition, our data reveal a clear relation between the *KRAS*/*BRAF* mutation status and sensitivity to BH3 mimetics. Overall, we observe that targeting BCL-XL and MCL-1 simultaneously has a dramatic synergy in CRC cell lines irrespective of CMS. This investigation provides more insight into the relationship between molecular phenotypes of CRCs and their sensitivity to BH3 mimetics.

## 2. Results

### 2.1. CMSs Exhibit Differential Sensitivity to BH3 Mimetics

To assess the sensitivity of CRC to BH3 mimetics, a panel of 19 CRC cell lines were treated with a titration of four BH3 mimetics. The cell lines represent the heterogeneity present in CRC and have previously been classified based on their molecular profile. First, we titrated single BH3 mimetics to determine the individual IC50 values per cell line (Table 1, Figure 1A–D and Appendix A). Subsequently, a combination of two mimetics were used in an 1:1 ratio to target different combinations of anti-apoptotic BCL-2 family members (Table 1, Figure 1A–D and Appendix A). Under the combinatory conditions, a concentration of 1 nM indicates 1 nM of mimetic 1 and 1 nM of mimetic 2 added together. Dose-response curves and IC50 values of the combinations were generated and calculated. Inhibitory effects of the combinatorial treatment at each concentration were calculated and plotted as dose-response curves (Figure 1A–D and Appendix A).

Generally, in single treatments, ABT-263 exhibited a relatively higher efficiency in impairing the viability of the majority of CRC lines compared to the other three BH3 mimetics (Figure 1A–I and Appendix A). This is likely due to the capacity of ABT-263 to target multiple anti-apoptotic family members (BCL-2, BCL-XL, and BCL-W) simultaneously (Table 1). In agreement, when single inhibitors targeting BCL-2 and BCL-XL were combined using the specific mimetics ABT-199 and A-1155463, a similar increase in efficacy was observed. The resultant IC50 values of this combinatorial treatment is strongly correlated to the IC50 values that were determined for the ABT-263 treatment (Appendix A). Importantly, as the combined inhibition of BCL-2 and BCL-XL with ABT-199/A-1155463 mirrors the efficacy of the BCL-2/BCL-XL/BCL-W inhibitor ABT-263, these data suggest that the role of BCL-W in protecting CRC cell lines is minimal (Appendix A).

Next, the relation between molecular features and BH3 mimetic sensitivity was analyzed. This revealed that CMS2 cell lines displayed increased resistance to ABT-199 and this was statistically significant when compared to CMS4 lines, while this trend was also observed in comparison to CMS1 and 3 (Figure 1F). Interestingly, CMS2 lines were also more resistant to MCL-1 inhibition by AZD5991, which was significant when compared with CMS3 and CMS4 (Figure 1H), indicating that MCL-1 was more critical in protecting CMS3 and CMS4. No CMS association was observed in A-1155463 and ABT-263 treated cell lines (Figure 1G,I).

The increased efficacy of ABT-263 in comparison to single inhibitors A-1155643 and ABT-199 indicated that CRCs utilize multiple anti-apoptotic family members to prevent the induction of apoptosis. Indeed, the analysis of distinct BH3 mimetic combinations revealed a dramatic synergy between combined BCL-XL and MCL-1 targeting with A-1155463/AZD5991 and/or ABT-263/AZD5991 (Figure 1A–E,J–M and Appendix A). In agreement, the IC50 value was consistently lower for the combination of MCL-1/BCL-XL targeting in relation to targeting of BCL-2/BCL-XL or BCL-2/MCL-1 (Figure 1E,J–L). To further verify the synergy and analyze the apoptotic impact exerted by the mimetics, a sub-lethal dose (1 μM) of each BH3 mimetic was administered alone or in combination in four CRC lines representative of each CMS and the percentage of cells with activated caspase-3 was measured by flow cytometry. Importantly, caspase activation was observed with combined inhibition of BCL-XL and MCL-1 (Figure 2A–D) and the level of caspase activity perfectly aligned with the IC50 values that were determined with Cell titer blue (CTB) (Figure 1E). For instance, ABT-263 + AZD5991 on KM12 was not effective in both CTB and caspase-3 activation, while the same combination effectively killed HT55 in both assays. Moreover, cell death induced by this synergistic combination could be completely blocked by pan-caspase inhibitor Q-VD-OPh suggesting that cell death was caspase dependent and predominantly apoptotic (Appendix A). Additionally, to assess if the cytotoxic effect of different BH3 mimetic combinations is synergistic or not, Bliss synergy scores were calculated for each combination in all CRC lines. This analysis further confirmed that A-1155463/AZD5991 is a highly synergistic combination in CRC (Figure 2E,F). Our data thus indicate that CRC cell lines are strongly dependent on BCL-XL and MCL-1, while the role of BCL-2 is less evident in most lines. Importantly, despite the enhanced efficacy of dual inhibitors, CMS2 lines remained relatively resistant, with especially the lines RCM-1 and SNU-C1 displaying relatively high IC50 values for the most effective combination (A-1155463/AZD5991). CMS2 lines also showed increased resistance to the combined inhibition of BCL-2 and MCL-1 (ABT-199/AZD5991) (Figure 1K), while no CMS association was observed in the combined inhibition of BCL-2 and BCL-XL (ABT-199/A-1155463) (Figure 1J). Taken together, different CMSs exhibited distinct responses to different BH3 mimetics, with especially the canonical CMS2 lines showing decreased sensitivity to BH3 mimetics.

### 2.2. Sensitivity to BH3 Mimetics Is Not Determined by Expression of the Individual Anti-Apoptotic BCL-2 Family Members

In order to reveal the underlying mechanisms determining differential sensitivity to BH3 mimetics in CRC, mRNA and protein expression of BCL-2, BCL-XL, and MCL-1 were examined in all 19 CRC cell lines in untreated conditions. Generally, *BCL-XL* and *MCL1* mRNA expression was more abundant than BCL-2, which was consistent with the drug screening data showing higher dependency on BCL-XL and MCL-1 in CRC cell lines (Figure 3A–G). For *BCL-2* and *BCL-XL*, mRNA expression levels correlated with the corresponding protein level (Figure 3H,I). The mRNA and protein levels of *MCL1* were clearly less well correlated (Figure 3C,J), which is likely due to the extensive post-translational regulation of protein stability described for this family member [15,16]. 

On average, there was no clear subtype-related *BCL-2* mRNA and protein expression (Figure 3A,D–G). In contrast, CMS4 cell lines appeared to have lower levels of BCL-XL protein (Figure 3B,D–G). Strikingly, MCL-1 expression was relatively high in the majority of CMS2 lines (Figure 3D–G), which related to their relative resistance to MCL-1 targeting. Combined with a relatively high average expression of BCL-XL, this explains why CMS2 lines were more refractory to (combined) BH3 mimetics. Despite these group-based patterns there was no direct association between target expression and sensitivity to the specific BH3 mimetics (Appendix A). Moreover, even when combining expression patterns of the three anti-apoptotic family members, it did not reveal a clear association between sensitivity and expression (data not shown). Conclusively, CRC cell lines show distinct anti-apoptotic BCL-2 proteins expression and depend on multiple anti-apoptotic BCL-2 proteins, which is not correlated to sensitivity to BH3 mimetics. 

### 2.3. Sensitivity to BH3 Mimetics Correlates with the KRAS/BRAF Mutation Status

Next to the association patterns between CMS and sensitivity, the relation between individual oncogenic mutations and sensitivity was analyzed. Previous findings have highlighted an important role for p53 signaling in the regulation of apoptosis, but also of the RAS/RAF/MAPK pathway in regulating sensitivity to mitochondrial dependent apoptosis. As these pathways are frequently mutated in CRC, we analyzed the association between BH3 mimetic response and specific molecular features. To this end, we grouped CRC cell lines based on their molecular features including microsatellite instability (MSI vs. MSS, Figure 4A and Appendix A), CpG island methylator phenotype (CIMP-high vs. CIMP-low, Figure 4B and Appendix A), and five most common gene mutations in CRC (APC, KRAS, BRAF, TP53, PIK3CA mutant vs. wild type, Figure 4C–G and Appendix A). 

This analysis revealed no significant correlation between the sensitivity and MSI or CIMP status of the cell lines (Figure 4A,B and Appendix A). Similarly, APC and PIK3CA mutations did not show a significant correlation with sensitivity (Figure 4C,G and Appendix A). Surprisingly, despite its clear role in apoptosis signaling, no relation between the p53 mutation status and sensitivity was found (Figure 4F and Appendix A). In contrast, KRAS-mutant CRC cells exhibited significantly less sensitivity to MCL-1 inhibition or BCL-2/MCL-1 co-inhibition (Figure 4D and Appendix A). Surprisingly, this relative resistance was not observed for the BRAF mutation, which is downstream of KRAS in the signaling pathway. If anything, the reverse was observed and indeed a significant higher sensitivity was evident for BCL-2 inhibition (Figure 4E), as well as several combinatory treatments (Appendix A). 

## 3. Discussion

Evasion of apoptosis is a hallmark of cancer, contributing to tumor cell survival and therapy resistance. Tumor cells often upregulate several anti-apoptotic BCL-2 family proteins as a survival mechanism and in CRC, we and others have previously shown that BCL-XL plays a crucial role, particularly also in the stem cell compartment [17,18]. However, we have recently shown that CRC tumors are highly heterogeneous and can be unbiasedly classified into four distinct subtypes, each with unique features and therapy response [4,14]. In this study, by examining anti-apoptotic dependencies in the context of these subtypes and other defining molecular features of CRC, we provide an in-depth overview of not only the therapeutic vulnerabilities but also the predictors of response to BH3 mimetics. 

When testing individual BH3 mimetics, we find that the inhibition of multiple BCL-2 family members by ABT-263 shows higher efficiency in CRC compared to BCL-2, BCL-XL or MCL-1 inhibition alone. Therefore, we also tested all four BH3 mimetics in combination and found that especially BCL-XL and MCL-1 inhibition is highly synergistic in all 19 CRC cell lines. Unsurprisingly, as competent inhibitors of anti-apoptotic proteins that are capable of inducing mitochondrial outer membrane permeabilization (MOMP), a combination of BH3 mimetics is sufficient to activate caspase-3 and their efficacy is dampened by the pan-caspase inhibitor Q-VD-OPh, which suggests that caspase-dependent apoptotic cell death is the predominant pathway induced upon the BH3 mimetic treatment. However, as reported by Tait et al. [19], cells can still die upon MOMP even in the absence of caspase activity via a so-called caspase independent cell death (CICD). Although this was not observed in the time-frame tested here, we cannot exclude that after a longer exposure time, BH3 mimetics treated cells may still succumb to the mitochondrial insult and hence undergo CICD after MOMP. Additionally, this does not rule out that other types of cell death that can also be blocked by Q-VD-Oph, such as pyroptosis, might be involved. In general, CRC cells express higher levels of *BCL-XL* and *MCL1*, while *BCL-2* mRNA levels are lower, suggesting that they are more dependent on the former two for apoptosis resistance. In line with our observation, Luo et al. have also shown that the co-inhibition of BCL-XL and MCL-1 using A-1331852 and S64845 showed higher cytotoxicity than the co-inhibition of BCL-XL/BCL-2 or MCL-1/BCL-2 [20]. Moreover, this potent combination has been shown to induce apoptosis in HCT-116 even in the absence of all BH3-only proteins, in a BAX-dependent manner [21]. A similar synergy between BCL-XL and MCL-1 inhibition has also been observed in other types of solid tumor such as cervical cancer [22] and melanoma [23]. Taken together, our data suggest that BCL-XL and MCL-1 could contribute complementarily to maintaining cancer cell survival and co-inhibition, therefore dramatically enhances cytotoxicity.

We have previously shown that CRC cell lines vary in their response to chemotherapy in vitro and in vivo, with CMS4 tumors showing increased resistance [14]. In this study, we also observe a differential response to the BH3 mimetic treatment depending on the subtype. With a single inhibitor treatment, we find that CMS2 cell lines are particularly resistant to BCL-2 and MCL-1 inhibition. This resistance is also observed in the combination therapy setting as CMS2 lines have higher IC50 values for most combinations of BH3 mimetics. This is a rather surprising observation in light of the observed sensitivity of CMS2 lines to chemotherapy. We did observe that CMS2 cell lines express higher levels of BCL-XL and MCL-1 in general, which might contribute to the observed resistance to BH3 mimetics. Importantly, all CMS4 lines tested harbor wild type *KRAS*, while the majority of CMS2 (4/6) cell lines have a *KRAS* mutation, which could explain the higher resistance to MCL-1 inhibition as suggested by the correlation analysis. Nevertheless, regardless of this basal resistance to MCL-1 inhibition, the combined inhibition of MCL-1 and BCL-XL also induces potent cell death in this subtype, as well.

The underlying reason for the observed difference between BH3 mimetic sensitivity of CMS4 lines in comparison to the reported resistance towards chemotherapy remains to be established. This is especially important to understand when related to the higher sensitivity of CMS2 tumors towards chemotherapy, while they present with lower sensitivity towards BH3 mimetics. This suggests that anti-apoptotic protein reliance on its own does not completely explain the difference in chemotherapy response between these subtypes. Obviously, chemotherapy sensitivity is defined by multiple aspects and does not only involve cell cycle speed, which is reportedly lower for CMS2 lines [14], but also involves the presence of downstream pathways and BH3 proteins signaling towards the mitochondria. Moreover, chemotherapy efficacy is also strongly dependent on drug efflux pumps, which are differentially expressed between the cell lines (results not shown). Therefore, direct relations between chemotherapy and BH3 sensitivity are difficult to draw. Nevertheless, our data do allow us to conclude that BH3 mimetics may provide a better option for mesenchymal CMS4 tumors as compared to chemotherapy, showing relatively effective cell death induction especially for the combination of BCL-XL and MCL-1 targeting mimetics. 

To further address if the differential sensitivity to BH3 mimetics could be defined by the corresponding target expression, BCL-2, BCL-XL, and MCL-1 protein levels were defined, which revealed that the sensitivity is neither determined by the relative expression of the corresponding targets of BH3 mimetics nor the overall expression of all three anti-apoptotic BCL-2 members. Consistently, Smith et al. also indicated that the co-expression of related anti-apoptotic BCL-2 family proteins may limit the activity of ABT-199 in diffuse large B cell lymphoma despite having a high BCL-2 expression [24]. On the other hand, Touzeau et al. have shown that sensitivity to BH3 mimetics in multiple myeloma could be related to BH3 profiling suggesting that the BH3 only protein expression is also involved in defining sensitivity to BH3 mimetics [25]. BCL-XL sensitivity has been shown to closely relate to MCL-1 activity, in particular predicted by NOXA levels, which specifically inhibits MCL-1 [26,27]. Furthermore, a recent study has used a computational model reflecting the dynamic regulation of this pathway in order to identify high-risk CRC patients [28]. Considering that apoptosis is tightly regulated by a balance of interactions between BCL-2 family members, it is likely that an overarching understanding of this balance in a context-specific manner is necessary to determine the BH3 mimetic response. Alternatively, a simple screen such as performed here or performed by BH3 profiling [29], will provide such an overarching snapshot of the sensitivity of cancers to different mimetics.

In addition to CMS and BH3 mimetic target expression, we also assessed if the mutation status influences the BH3 mimetic response. This analysis revealed that CRC cell lines harboring KRAS mutations are less responsive to MCL-1 inhibition than the wild type lines, indicating that KRAS activation might be involved in resistance to apoptosis mediated by MCL-1. It has been reported by Okamoto et al. that CRC tumors with mutant KRAS are able to upregulate BCL-XL via the ERK pathway, which confers resistance to the proteasome inhibitor, Carfilzomib [30]. However, it is unexpected that BRAF-mutant lines have higher sensitivity to BCL-2 inhibition. It has been shown that BRAF mutation in melanoma could increase MCL-1 expression, suggesting that activating BRAF mutations would confer resistance to apoptosis [31]. Furthermore, it is surprising that TP53 mutations show no correlation with the efficacy of BH3 mimetics, considering the crucial role that TP53 plays in apoptosis regulation [32]. Therefore, a further investigation is required to gain a comprehensive understanding of the exact implications of these oncogenic mutations in determining susceptibility to BH3 mimetics. 

Altogether, we show that CRC cell lines display differential response to BH3 mimetics, which in part relates to CMS and KRAS or BRAF mutation status. Our findings provide an in-depth overview of BCL-2 family expression and sensitivity in the context of several molecular features, which can guide future investigations to optimize the application of BH3 mimetics in CRC. Furthermore, our results emphasize the potent efficacy of combined MCL1 and BCL-XL inhibition for CRC therapy, regardless of pre-existing molecular features. 

## 4. Materials and Methods

### 4.1. Cell Lines and Profiling

Nineteen colorectal cancer cell lines were kindly provided by Sanger Institute (Cambridge, UK) and authenticated by STR profiling. Cell lines RKO, SW48, HT55, SW948, T84, CL-40, LS-180, CaR-1, HUTU-80, and OUMS-23 were maintained in Dulbecco’s modified Eagle’s medium/F-12 medium with L-glutamine, 15 mM HEPES (Gibco, Paisley, Scotland) supplemented with 8% fetal bovine serum (Serana, Pessin, Germany) and 50 units/mL of penicillin and streptomycin. HCT-116, KM12, RCM-1, SNU-C1, LS-1034, LS-513, COLO-320-HSR, MDST8, and NCI-H716 were maintained in RPMI 1640 with L-glutamine, 25 mM HEPES (Gibco, Paisley, Scotland) supplemented with 8% fetal bovine serum, 50 units/mL of penicillin and streptomycin, 1% D-glucose solution plus (Sigma-Aldrich, Saint Louis, MO, USA), and 100 μM sodium pyruvate (Gibco, Paisley, Scotland). All cells were cultured in a humidified atmosphere at 37 °C 5% CO_2_.

All 19 CRC lines were classified into four CMSs by our previous work based on the consensus molecular pattern described in [14]. The gene mutation status, microsatellite instability (MSI), and CpG island methylator phenotype (CIMP) of all these 19 lines were also determined as shown in [14]. 

### 4.2. Compounds

ABT-199, ABT-263, and Q-VD-OPh were purchased from Selleckchem, Houston, TX, US. A-1155463 and AZD5991 were purchased from Chemietek, Indianapolis, IN, US. All compounds were dissolved in dimethyl sulfoxide (DMSO) at a 10 mM stock concentration.

### 4.3. Cell Viability Assay

To assess the sensitivity to four BH3 mimetics, CRC cells were plated into 96-well plates and treated with two BH3 mimetics in a matrix dilution at a time. After 48 h of treatment, the CellTiter-Blue^®^ cell viability assay (Promega, Madison, WI, USA) was used to determine the cell viability by measuring mitochondrial respiration according to the manufacturer’s instruction. The relative viability was calculated by normalizing to the untreated control. Dose-response curves and IC50 values were generated and calculated on GraphPad Prism 8.0 (GSL Biotech LLC, Biomatters, Chicago, IL, USA) with log(inhibitor) vs. normalized response—variable slope.

### 4.4. Flow Cytometry and CaspaTag Staining 

Fifty thousand cells were plated and treated with different BH3 mimetics. After 48 h, cells were trypsinized and harvested. CaspaTag caspase-3/7 in situ assay kit, Sulforhodamine (Merck, Sigma-Aldrich, USA) or propidium iodide (Sigma-Aldrich, Saint Louis, MO, USA) were used to label cells to detect activated caspase-3 or compromised cytoplasmic membrane, respectively. The percentage of positive cells were measured by flow cytometry on the CytoFLEX (Beckman Coulter Life Sciences, Brea, CA, USA). 

### 4.5. Bliss Synergy Scoring

According to [33], the Bliss Synergy score was automatically calculated by Synergyfinder [34]. To interpret the results, scores below −10 were considered antagonistic. Scores between −10 and 10 were considered additive and scores above 10 were considered synergistic.

### 4.6. The mRNA Expression Analysis

The RNA expression analysis of the CRC cell lines was performed as described in [14]. Briefly, microarrays expression profiles were obtained using the GeneTitantm MC system from Affymetrix according to the standard protocols of the Cologne Center for Genomics (CCG), University of Cologne, Germany. The dataset is publicly available in the gene expression omnibus (GEO) repository under the accession number GSE100478. Data were normalized using the robust multi-array average (rma) method as implemented in the affy R package (version 1.52.0). Probes were annotated with the hgu133plus2.db R data package (version 3.2.3).

### 4.7. Western Blotting

For the Western blot analysis of BCL-2 family members, cells were lysed using the 1× RIPA lysis and extraction buffer (Thermo Fischer Scientific, Waltham, MA, USA) containing Halt protease and phosphatase inhibitor cocktail (1:100, Thermo Fischer Scientific, Waltham, MA, USA). Protein samples were quantified using the Pierce BCA protein assay kit (Thermo Fischer Scientific, Waltham, MA, USA) as per the manufacturer’s instructions. In addition, 20 µg of protein was loaded per well into 4–15% precast gels (Bio-Rad, Hercules, CA, USA) and then transferred to PVDF membranes using the Trans-Blot Turbo transfer system (Bio-Rad, Hercules, CA, USA) according to the manufacturer’s instructions using the mixed molecular weight transfer settings. Membranes were blocked for 1 h in 5% bovine serum albumin (BSA) in Tris-buffered saline and Tween-20 (TBS-T,1×) and stained with a primary antibody overnight at 4 °C. The following primary antibodies were tested: BCL-2 (1:1000 #15071, Cell Signaling, Danvers, MA. USA), BCL-XL (1:1000, #2764, Cell Signaling, Danvers, MA, USA), and MCL-1 (1:1000, #4572, Cell Signaling, Danvers, MA, USA), all diluted in 5% BSA in TBS-T. After washing the blots four times for 20 min each with TBS-T, the secondary antibody anti-rabbit-horseradish peroxidase (1:5000, #4050-05, Southern Biotech, Birmingham, AL, USA) or anti-mouse-horseradish peroxidase (1:5000, #1031-05, Southern Biotech, Birmingham, AL, USA) was added for 2 h at room temperature. Following another round of 4 × 20 min washes, the membranes were developed using the LumiLight Western blotting substrate (Sigma-Aldrich, Saint Louis, MO, USA) and imaged on the ImageQuant LAS4000 (GE Healthcare Life Sciences, Chicago, IL, USA). Before proceeding with blotting for protein expression, the polyacrylamide gel was incubated for 5 min in an electrophoresis buffer containing 1% 2,2,2-Trichloroethanol (2,2,2-TCE, cat. #T54801, Sigma-Aldrich) to allow for tryptophan visualization and thereby a comparison of the amount of protein loaded between the cell lines [35]. The gels were imaged using the UV sample tray of a ChemiDoc imaging system (Bio-Rad, Hercules, CA, USA). Western blot images were quantified using ImageJ, wherein a region of interest (ROI) was defined and used for each lane to select the band of interest. The mean grey value of the ROI was measured for each band as well as the background region of each lane. Each protein band expression was blank corrected and normalized to the 2′2′2′-TCE loading control levels.

### 4.8. Correlation Analysis and Statistics

To analyze the correlation between the drug sensitivity and the phenotype of CRC cell lines, IC50 values were plotted and grouped based on their CMSs or mutation status. The original tables of IC50 values and corresponding mutation status can be found in Appendix A. The significant difference between groups were tested by the one-tailed Mann-Whitney test. A *p* < 0.05 was considered statistically significant. The correlation between IC50 values and target expression were analyzed by Spearman’s rank correlation. 

## Figures and Tables

**Figure 1 ijms-22-03811-f001:**
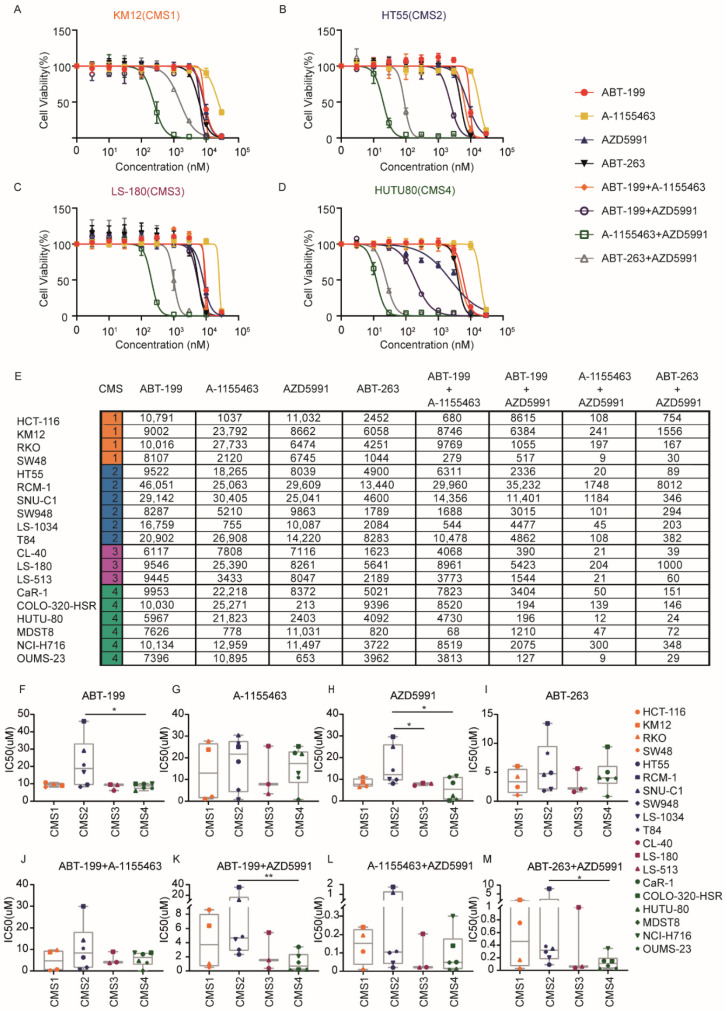
Differential sensitivity to BH3 mimetics in colorectal cancer (CRC) cells. (**A**–**D**) Dose-response curves of four BH3 mimetics and four combinations (1:1) in four representative CRC cell lines of each subtype. The Y-axis indicates the relative cell viability normalized to the untreated control. The legend for (**A**–**D**) indicates the symbol for the different mimetic treatments used. (**E**) IC50 values for BH3 mimetics and combinations tested on all 19 CRC cell lines, in nM; (**F**–**I**) IC50 values of four single BH3 mimetics in the CRC cell line panel grouped according to their CMS. *: *p* < 0,05; (**J**–**M**) IC50 values of four combinations (1:1) in the CRC cell line panel grouped according to their CMS. **: *p* < 0.01; *: *p* < 0,05, Mann-Whitney test (one-tailed). The legend indicates the symbols assigned for each cell line.

**Figure 2 ijms-22-03811-f002:**
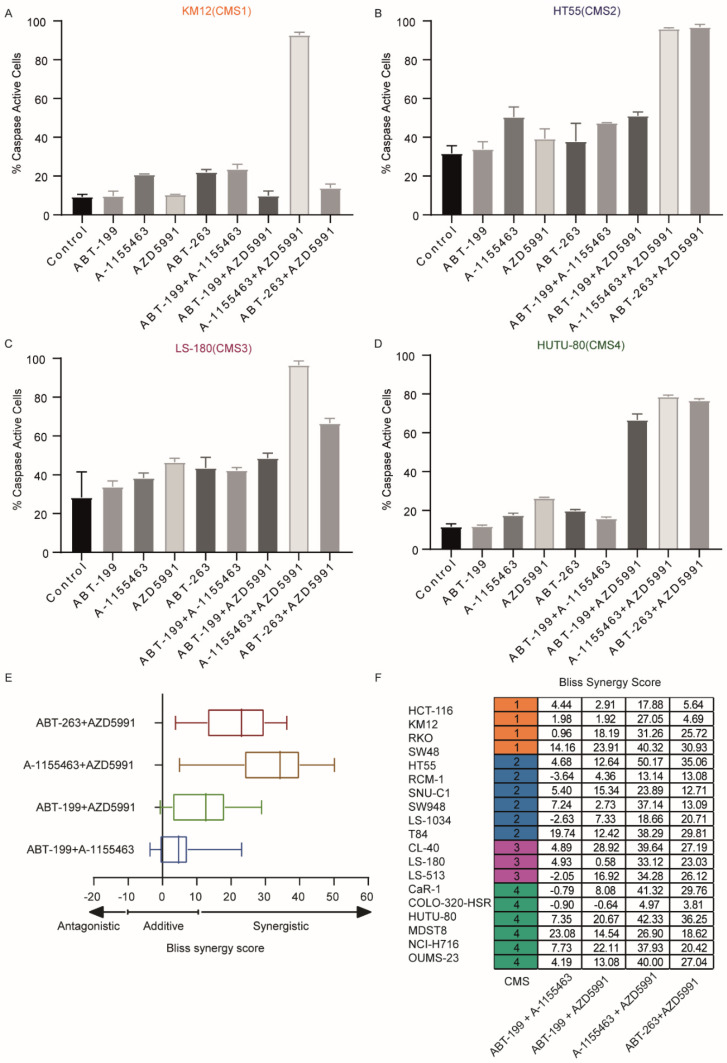
Synergy between different BH3 mimetics in CRC cells. (**A**–**D**) Flow cytometry analysis of the activation of caspase-3 induced by different BH3 mimetics and combinations in four representative CRC lines; (**E**,**F**) bliss synergy score of the four combinations for all CRC cell lines. Scores below −10 are considered antagonistic, while scores between −10 and 10 are considered additive. Scores above 10 are considered synergistic.

**Figure 3 ijms-22-03811-f003:**
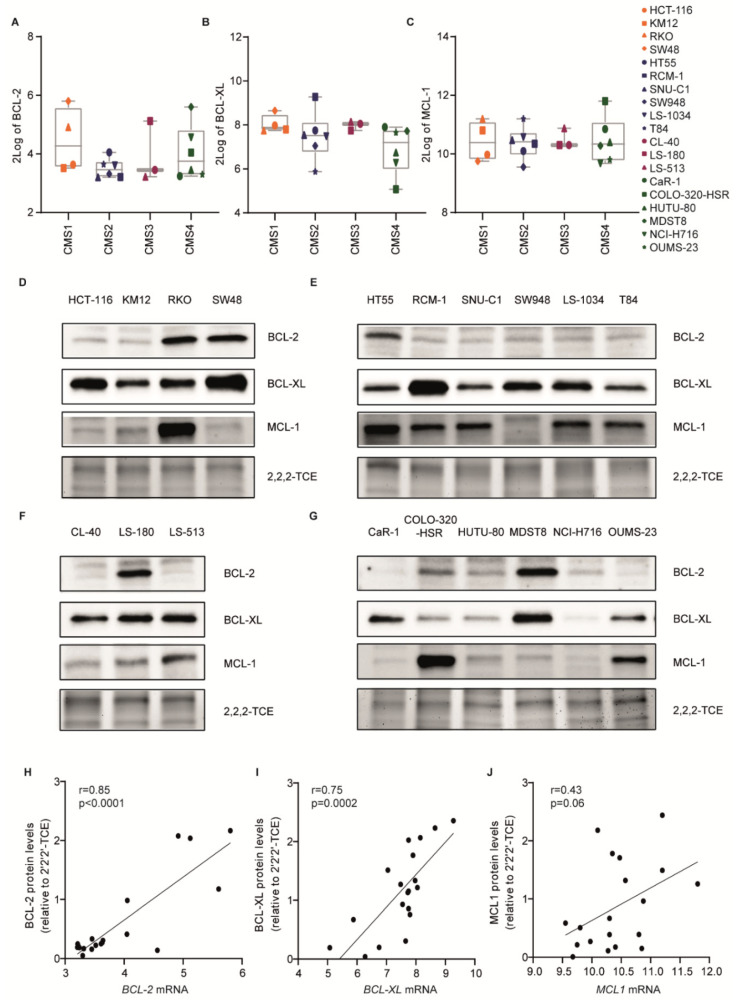
The mRNA and protein expression of BCL-2 family members in CRC cell lines. (**A**–**C**) 2Log expression of *BCL-2* (**A**), *BCL-XL* (**B**), and *MCL1* (**C**) mRNA in the CRC cell line panel grouped according to their CMS subtype. The legend indicates the symbols assigned for each cell line. (**D**–**G**) Western blot analysis for BCL-2, BCL-XL, and MCL-1 expression in CMS1 (**D**), CMS2 (**E**), CMS3 (**F**), and CMS4 (**G**) cell lines. 2,2,2-Trichloroethanol (2,2,2-TCE) signal (excerpt taken around 40 kDa region) indicates the amount of protein loaded per cell line. (**H**–**J**) Correlation between RNA (as in panel **A**–**C**) and protein (as in panel **D**–**G**) levels of BCL-2 (**H**), BCL-XL (**I**), and MCL-1 (**J**) in the cell line panel. N = 19. Pearson’s correlation (two-tailed *p*-value).

**Figure 4 ijms-22-03811-f004:**
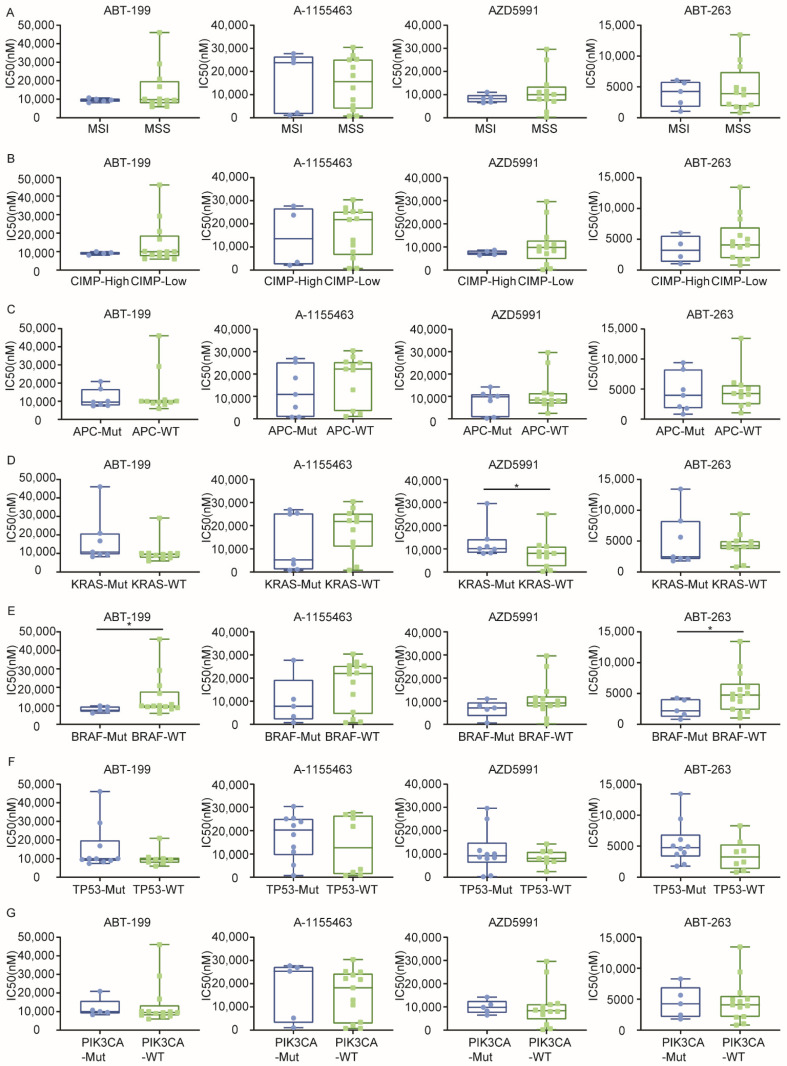
Correlation analysis of IC50 values of four BH3 mimetics and molecular phenotypes in CRCs. (**A**) Microsatellite instability MSI, MSI microsatellite instable; MSS: Microsatellite stable; (**B**) CpG island methylator phenotype CIMP, CIMP-high: High CpG island methylation; CIMP-low: Low CpG island methylation; (**C**–**G**) Mutation: Mut: Mutant; WT: Wild type; *: *p* < 0,05; Mann-Whitney test (one-tailed).

**Table 1 ijms-22-03811-t001:** Overview of the targets of the inhibitors and combinations.

Inhibitors and Combinations	BCL-2	BCL-XL	MCL-1	BCL-W
ABT-199	++	−	−	−
A-1155463	−	++	−	−
AZD5991	−	−	++	−
ABT-263	+	+	−	+
ABT-199 + A-1155463	++	++	−	−
ABT-199 + AZD5991	++	−	++	−
A-1155463 + AZD5991	−	++	++	−
ABT-263 + AZD5991	+	+	++	+

## Data Availability

Microarrays expression profiles were obtained using the GeneTitantm MC system from Affymetrix according to the standard protocols of the Cologne Center for Genomics (CCG), University of Cologne, Germany. The dataset is publically available in the Gene Expression Omnibus (GEO) repository under the accession number GSE100478. Clinical trial data is available on https://clinicaltrials.gov/ and accessed by following references: ClinicalTrials.gov Identifier: NCT03218683 first posted on 14 July 2017; NCT02591095 first posted on 29 October 2015; NCT00878449 first posted on 9 April 2009; NCT00891605 on 1 May 2009.

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
