# Peer review of "BH3 Mimetic Sensitivity of Colorectal Cancer Cell Lines in Correlation with Molecular Features Identifies Predictors of Response"

_ijms, 2021, doi:10.3390/ijms22083811_

Round 1
Reviewer 1 Report
The study titled "BH3 mimetic sensitivity of colorectal cancer cell lines in correlation with molecular features identifies predictors of response" is adequate for publication.
Are there different pathways of cell death activated by the various combination compounds apart from apoptosis (this should be discussed in the discussion part)
Author Response
- The study titled "BH3 mimetic sensitivity of colorectal cancer cell lines in correlation with molecular features identifies predictors of response" is adequate for publication.
We would like to thank the reviewer for these kind comments.
- Are there different pathways of cell death activated by the various combination compounds apart from apoptosis (this should be discussed in the discussion part).
This is an important point. BH3 mimetics are of course designed to activate the mitochondrial pathway to apoptosis. In the revised manuscript we have added a figure (new figure 2) to support the activation of this pathway using one dose of either single or combination treatments with mimetics. We used caspase 3 activity as a measure for downstream pathway activation. Moreover, we used a pan-caspase inhibitor to prevent the activity of caspases altogether and tested the impact on BH3 mimetic-induced death. We observed that most effective combination of A-1155463/AZD5991 dramatically induced caspase 3 activation in 4 representative CRC lines. Apoptotic cell death can be blocked by pan caspase inhibitor Q-VD-OPh suggesting the BH3 mimetics induced death is predominantly apoptotic. However, our observation is not in agreement with the observation from Stephen Tait’s group who indicated MOMP induced cell death is caspase independent as they have shown that caspase inhibitor didn’t rescue the cell from MOMP induced cell death, which was not the case in our observation. The possible explanation could be that if we carry the treatment on for longer, the caspase inhibitor might no longer be protective and cells would die from caspase-independent cell death. To discuss these data, we have also added a small section to the discussion. See Line 244-255.
Reviewer 2 Report
The study highlights the effectiveness of BH3 mimetics in combinative therapy on a comprehensive number of CRC cell-lines which have been grouped based on heterogeneity. They also measured the mRNA and protein levels in BCL-XL, MCL-1 and Bcl-2 in the 19 CRC celllines. Further analysis of the molecular features in relation to dose-response found significant resistance in cell-lines which are wildtype BRAF to ABT-264 or AZD5991 when KRAS mutated. The promising findings are the synergic effectveness of A1155463 and AZD5991 in all CRC cell-lines tested.
On Line 42, CMS4 has the poor prognosis with ‘chemotherapy resistance’. However, in figure 1 and supplementary figure 2, the wide deviation in IC50 values is shown in CMS2 suggesting resistance for all the single treatments and the IC50 values are much higher even in combination compared to CMS4. Please explain.
On line 209, the authors mention ‘CMS4 tumors showing increased resistance’ but in figure 1G and 2H, there is a significant difference between IC50 for CMS2 and CMS4, but also CMS2 and CMS3. A similar significant difference in figure 1K and 1M for combinative treatments indicates a higher IC50 in CMS2 compared to CMS4
On line 86, you suggest ‘the role of BCL-W in protecting CRC lines is minimal’ based on supplementary figure 1. Can you please further elaborate how the significant trend between a combinative and singular treatment suggests this?
On Line 194, the authors mention ‘ABT-263 shows higher efficiency in CRC compared to BCL-2, BCL-XL or MCL-1 inhibition alone.’ But in figure 1E, AZD5991 which only inhibits MCL-1 has a lower IC50 value in COLO-320-HSR, HUTU-80 and OUMS-23. Can you please clarify this statement.
For supplementary figure 1, can you explain which IC50 values the y-axis is plotted against.
Figure 2D-G, please clarify the treatment used and concentration.
For figure 2D-G, can you please provide a control treatment group to demonstrate the initial level of BCL2, BCL-XL and MCL-1 of each cell-line before adding mimetic.
For figure 2D-2G, can you provide evidence to normalize the protein level between the cell-lines in the form of a loading control e.g., GAPDH.
In Figure 1, can you please clarify the representative mimetic to the corresponding IC50 values for each combination treatment.
For figure 1F-M, 2A-C and 3, can you please provide a key to differentiate between the cell-lines in each graph.
In the introduction, please provide more background on the mimetics which the study is focused upon.
Can you please provide additional data to verify the effects of combinative and singular treatment on the CRC cell-lines?
Can you please clarify whether the interaction of the mimetics in each combinative treatment is synergistic, additive, or antagonistic?
Author Response
1. On Line 42, CMS4 has the poor prognosis with ‘chemotherapy resistance’. However, in figure 1 and supplementary figure 2, the wide deviation in IC50 values is shown in CMS2 suggesting resistance for all the single treatments and the IC50 values are much higher even in combination compared to CMS4. Please explain.
2. On line 209, the authors mention ‘CMS4 tumors showing increased resistance’ but in figure 1G and 2H, there is a significant difference between IC50 for CMS2 and CMS4, but also CMS2 and CMS3. A similar significant difference in figure 1K and 1M for combinative treatments indicates a higher IC50 in CMS2 compared to CMS4
These are both crucial observations and valid points raised by the reviewer. In the original manuscript we had discussed this discrepancy. In the revised version, we have extended this discussion and provided potential explanations for this discrepancy. Indeed, CMS4 sensitivity towards BH3 but not chemotherapy is an important finding of our study but vice versa, chemotherapy sensitivity and higher resistance to BH3 mimetics for CMS2 is surprising. We do not have data to explain the difference but have discussed the potential role of mutations as well as drug efflux pumps. See Line 274-276 and 280-294.
3. On line 86, you suggest ‘the role of BCL-W in protecting CRC lines is minimal’ based on supplementary figure 1. Can you please further elaborate how the significant trend between a combinative and singular treatment suggests this?
This is an important point and we apologize for not making this stand out clearer. The inhibition profile of ABT-263 is now indicated clearly in the text and table 1. It targets BCL-2 and BCL-XL but also BCL-W. ABT-199 is selective for BCL-2, while A-1155463 is selective for BCL-XL. As we combined ABT-199 and A-1155463 this condition can be easily compared to ABT-263.
ABT-263 targets three members, while ABT-199/A-115 targets only two of these and leaves BCL-W free to exert its anti-apoptotic function. Nevertheless, we find that the concentrations needed are very similar for both treatments. This suggests that targeting BCl-2 and BCL-XL gives the same efficacy as targeting all three members and hence indicates that the role for BCL-W is minimal in CRC protection.
To make this clearer we have added a bit of text on the selectivity of the compounds and related this to table 1 where this is listed as well. In addition, we have rewritten the text around Line 108-115.
4. On Line 194, the authors mention ‘ABT-263 shows higher efficiency in CRC compared to BCL-2, BCL-XL or MCL-1 inhibition alone.’ But in figure 1E, AZD5991 which only inhibits MCL-1 has a lower IC50 value in COLO-320-HSR, HUTU-80 and OUMS-23. Can you please clarify this statement.
Thank you so much for pointing this out and we apologize for not describing this observation precisely. We have rephrased this sentence to read “Generally, in single treatments, ABT-263 exhibited relatively higher efficiency in impairing the viability of majority of CRC lines compared to the other three BH3 mimetics (Figure 1A-I and Supplementary Figure 2A)” (Line 111-113). The problem in directly comparing the compounds is that they do not all have the same affinity for their targets with ABT-263 clearly showing lower affinity for BCL-XL than the selective compound A-1155463. Therefore, in some cases A-1155463 alone is more effective than ABT-263. In addition, some tumors show stronger dependency on MCL-1. Nevertheless, in general dual targeting is more effective than single BCL-2 family member targeting, which is especially the case for BCL-XL/MCL-1 targeting.
We have changed the text to on one hand correct the statement and on the other to make the point stand out better. See Line 105 and 106.
5. For supplementary figure 1, can you explain which IC50 values the y-axis is plotted against.
Sorry that this is not clear. In supplementary figure 1 we plotted on the X-axis the IC50 value of ABT-263 per cell line (each dot is a cell line). The true values are shown in the table in figure 1. On the Y-axis we listed the IC50 of the combination treatment with A-1155463 and ABT-199. As indicated in the text now more clearly on page 2/3, the latter IC50 is determined by treating cells with an equimolar range of two compounds. So, mimetics are mixed at 1:1 ratio on the basis of molar concentration and therefore 1nM of mimetic combination indicates 1nM of A-1155463 and 1nM of ABT-199. This allows us to plot on the Y-axis the IC50 values of the joint inhibition of BCL-XL and BCL-2, while on the X-axis we plotted the IC50 of one mimetic that targets both BCL-2 and BCL-XL (ABT-163). The only difference between the two conditions is now the fact that ABT-263 also targets BCL-W.
As we see an almost perfect correlation between these two values, we also conclude that the role of BCL-W is relatively minor.
6. Figure 2D-G, please clarify the treatment used and concentration.
Sorry to see that this was not clear from the text. The mRNA and protein levels indicated represent basal expression levels. So the expression of BCL-2 family members without treatment. This is the proper measure as this is what the mimetics are up against when one treats the cells. Potential changes during treatment may be of interest but not topic of this study. We have changed the text here to make this more evident, see Line 171.
7. For figure 2D-G, can you please provide a control treatment group to demonstrate the initial level of BCL2, BCL-XL and MCL-1 of each cell-line before adding mimetic.
The reviewer raises an important point. Westerns require an internal control to guarantee equal loading. However, in our studies with these cell lines we have come to the conclusion that a good control that can be used to normalize along a vast range of cell lines is impossible to identify. GAPDH for instance shows more than 2-fold expression difference on mRNA and on protein level, this seems to be even more divergent. The same holds for Actin or ERK1/2, two other control antibodies that we routinely use. We have spent quite some time on trying to find proteins that could be used for normalization and have concluded that the best means to normalize is to use a generic protein quantification. 2-2-2- Trichloroethanol binds effectively to tryptophans (Ladner, C.L., et al., Visible fluorescent detection of proteins in polyacrylamide gels without staining. Anal Biochem, 2004. 326(1): p. 13-20.) and hence provides a perfect measure for the total protein quantity present on the blot. We have therefore used this to normalize, see Line 400-403.
8. In Figure 1, can you please clarify the representative mimetic to the corresponding IC50 values for each combination treatment.
Hopefully this has now been clarified under point 2 and 4. When IC50s are listed for combination treatments this indicates that both compounds are added to this condition at the indicated concentration. In other words when the IC50 is 1000nM this means the half maximal cell death observed occurs when both mimetic 1 and mimetic 2 are both dosed at 1000nM, see Line 95-100.
9. For figure 1F-M, 2A-C and 3, can you please provide a key to differentiate between the cell-lines in each graph.
For figure 1F-M and 2A-C, we changed the symbols and assigned a certain color symbol to visualize the cell lines in the graph. This is not working well in figure 3 as symbols pile up and colors make the figure rather less legible than better. Therefore, we have added the representative tables to the supplemental data. In each table the way cell lines are segregated into mutant of wild type or CIMPhigh/low MSI/MSS is indicated. This is a lot more convenient to go through and has been uploaded in sup table “Mutation and IC50 value tables.xlsx”.
10. In the introduction, please provide more background on the mimetics which the study is focused
We extended the introduction as requested, see Line 61-79.
11. Can you please provide additional data to verify the effects of combinative and singular treatment on the CRC cell-lines?
To address this point, we performed caspase 3 activity assays by FACS. This further supports the apoptotic effects of combinations and it indeed verified the combination of A-1155463/AZD5991 is most effective to induce apoptosis in four representative CRC lines, as shown in new Figure 2A-D. See Line 131-142.
12. Can you please clarify whether the interaction of the mimetics in each combinative treatment is synergistic, additive, or antagonistic?
Thank you for this important suggestion, we have added Bliss synergy scores and showed that the A-1155463/AZD5991 combination had a score range for most cell lines that is above 10, which indicates synergy. The combination of ABT-199+A1155463 showed scores from -10 to 10 which suggests additive effects of this combination. Combination of ABT-263 with AZD5991 and ABT199 with AZD5991 show the expected mixed outcomes. These data are all shown in new Figure 2E and F. See Line 138-142.
Reviewer 3 Report
Zhang et al. investigated the potential of BH3 mimetics to induce apoptosis in colorectal cancer cell lines having various consensus molecular subtypes.
Personally, I like the paper because it does not try to be more than what it is. It is a correct and well-done series of experiments comparing the IC50 values of several BH3 mimetics and their combinations in 19 CRC cell lines. The authors then try to gain a better understanding of their findings by considering the molecular profile of these cell lines (BCL-2, BCL-XL, MCL-1 levels along with KRAS, BRAF, TP53, PIK3CA status). Despite clear conclusions and correlations are difficult to make between these drugs and genotypes, it is nevertheless a valuable set of data that will be useful for future research. I recommend the manuscript for publication.
Minor points:
- I recommend the authors to revise their English language and style throughout the manuscript, which needs some polishing.
- The imager is called ImageQuant properly (not ImageQuanti).
Author Response
- Personally, I like the paper because it does not try to be more than what it is. It is a correct and well-done series of experiments comparing the IC50 values of several BH3 mimetics and their combinations in 19 CRC cell lines.
We thank the reviewer for these supportive words. We agree that the impact of our study is in the overview of the impact of all mimetics and combinations on a broader range of lines. Hopefully these data will provide a useful repository for other scientists in the research area.
- Minor points: I recommend the authors to revise their English language and style throughout the manuscript, which needs some polishing. The imager is called ImageQuant properly (not ImageQuanti).
We are sorry for the typos and sometimes convoluted wording. We have gone through the text several times and have significantly changed sections to improve clarity as well as style.
Round 2
Reviewer 2 Report
The authors have provided additional data to further verify the synergistic and apoptotic effect of the combinative treatments and hence support their conclusion.
A suggestion for the authors would be to provide dot plots or fluorescence intensity measurements of the controls, single and combinative treatments in Figure 2, to provide representation of the Caspase 3 flow cytometry data.